

# 2FAST2Q: a general-purpose sequence search and counting program for FASTQ files

Afonso M. Bravo[1], Athanasios Typas[2] and Jan-Willem Veening[1]

[1] Department of Fundamental Microbiology, University of Lausanne, Lausanne, Switzerland
[2] Genome Biology Unit, EMBL, Heidelberg, Germany

## ABSTRACT

**Background**. The increasingly widespread use of next generation sequencing protocols has brought the need for the development of user-friendly raw data processing tools. Here, we explore 2FAST2Q, a versatile and intuitive standalone program capable of extracting and counting feature occurrences in FASTQ files. Despite 2FAST2Q being previously described as part of a CRISPRi-seq analysis pipeline, in here we further elaborate on the program's functionality, and its broader applicability and functions.
**Methods**. 2FAST2Q is built in Python, with published standalone executables in Windows MS, MacOS, and Linux. It has a familiar user interface, and uses an advanced custom sequence searching algorithm.
**Results**. Using published CRISPRi datasets in which *Escherichia coli* and *Mycobacterium tuberculosis* gene essentiality, as well as host-cell sensitivity towards SARS-CoV2 infectivity were tested, we demonstrate that 2FAST2Q efficiently recapitulates published output in read counts per provided feature. We further show that 2FAST2Q can be used in any experimental setup that requires feature extraction from raw reads, being able to quickly handle Hamming distance based mismatch alignments, nucleotide wise Phred score filtering, custom read trimming, and sequence searching within a single program. Moreover, we exemplify how different FASTQ read filtering parameters impact downstream analysis, and suggest a default usage protocol. 2FAST2Q is easier to use and faster than currently available tools, efficiently processing not only CRISPRi-seq / random-barcode sequencing datasets on any up-to-date laptop, but also handling the advanced extraction of *de novo* features from FASTQ files. We expect that 2FAST2Q will not only be useful for people working in microbiology but also for other fields in which amplicon sequencing data is generated. 2FAST2Q is available as an executable file for all current operating systems without installation and as a Python3 module on the PyPI repository (available at https://veeninglab.com/2fast2q).

Corresponding authors
Afonso M. Bravo,
afonso.martinsbravo.1@unil.ch
Jan-Willem Veening, jan-willem.veening@unil.ch

## INTRODUCTION

Next generation sequencing (NGS) has drastically changed the landscape of experimental biology, not only by helping to characterize cellular networks to an unprecedented level, but also by generating vast quantities of data. Typical NGS data generated by Illumina

sequencing is delivered in the form of a so-called FASTQ file: a text file that contains the inferred DNA sequences with their respective quality scores, typically existing in a compressed form with the extension *.fastq.gz. However, as newer sequencing platforms become available, so do the sequencing file types. For example, Oxford Nanopore sequencers store their data in FAST5 format, which needs to be converted to FASTQ before any traditional downstream sequence analysis can be performed. More than the file itself, the compression format can also vary: DRAGEN ORA (.ora) is currently being rolled out by Illumina as an alternative to the standard .gz format. Despite these constant advancements, FASTQ remains the standard format, in large part probably due to the current convergence of NGS analysis programs to mainly accept FASTQ as first input. In time, however, format complexity might increase and lead to the requirement of further pre-analysis format-exchange programs, or the rewrite of current bioinformatics core programs.

Since big data analysis becomes an increasingly needed skill in biology, the demand for versatile user-friendly applications also rises. As NGS becomes simpler and widespread, so must its respective data processing. Thus, there is a need for intuitive, reproducible, and versatile tools that can handle the sometimes overwhelming initial raw data processing step.

NGS applications often require features to be extracted and counted from FASTQ files for downstream analysis. Several analysis tools and scripts exist for systematic reverse genetic screens, such as CRISPRi-seq (*Liu et al., 2021*) and random-barcode sequencing (RB-Seq) (*Wetmore et al., 2015*; *Cain et al., 2020*). At the moment, such pipelines tend to overspecialize into CRISPR/Cas9 workflows, are complex, or require informatics skills beyond the average user (*Winter et al., 2017*; *Li et al., 2014*; *Liao, Smyth & Shi, 2019*; *Winter et al., 2016*). A notable example, MAGeCK, allows for both feature counting and downstream feature differential analysis (*Li et al., 2014*). Due to being primarily optimized for the latter, it has some caveats regarding more complex feature extractions procedures. Indeed, when dealing with mismatches or dynamic read trimming/feature extraction it requires the installation of 3rd party command line only software such as bowtie2 and/or cutadapt. Current, more user friendly approaches such as CRISPRAnalyzeR and PinAPL-Py are also limiting in throughput in regards to searching and returning reads with specific sequences, especially when considering sequence mismatches, nucleotide wise Phred score filtering, and dynamic sequence search using multiple sequences of variable length (*Winter et al., 2017*; *Spahn et al., 2017*). This is particularly important in the cases where a user wants to control these sequencing processing parameters to analyze their experimental setup. As such, when these advanced requirements are needed, it is the current standard to create custom made pipelines for handling the specifics of the experiment, normally in conjunction with bioinformatics tools such as Trimmomatic, cutadapt, and/or Bowtie2 (*Bolger, Lohse & Usadel, 2014*; *Langmead & Salzberg, 2012*; *Martin, 2011*). When handling highly variable, and/or trying to retrieve unknown sequences from NGS data, the user will thus have no choice but to create their own custom made processing scripts, or outsource the data processing. The first assumes bioinformatics, sequencing, and programming knowledge,

requiring weeks or months of time to implement from scratch for the average user, while the latter needs either extra funds or the right willing colleague.

Here, we explore 2FAST2Q, a fast and versatile FASTQ file processor for extracting and counting sequence occurrences from raw reads. 2FAST2Q requires no installation by default, and works in all common operative systems. 2FAST2Q has been previously published as part of a CRISPRi-seq protocol, however, in this work we further elaborate on the program's functionalities (*De Bakker et al., 2022*). We demonstrate novel applications and provide an in-depth description of 2FAST2Q. As a proof of concept, we show that 2FAST2Q efficiently and reliably counts single guide RNA (sgRNA) features in FASTQ files originating from published prokaryotic and eukaryotic CRISPRi-seq experiments. Moreover, we explore 2FAST2Q novel functions, and how these can be used for any *de novo* sequence searching, or for extracting and counting any kind of sequences from FASTQ files using advanced search and filtering methods.

## MATERIALS & METHODS

### Installation and code availability

All 2FAST2Q executable files can be downloaded from zenodo: https://zenodo.org/record/5410822. The code, usage instructions, and test datasets are available on GitHub: https://github.com/veeninglab/2FAST2Q. 2FAST2Q is also a Python package, and can be accessed on PyPI: https://pypi.org/project/fast2q/. When using the executable version on MS Windows or MacOS, no further installation is required and a double click on the executable should suffice. For a more in depth description, please see the online tutorial on https://veeninglab.com/2fast2q. 2FAST2Q is fully implemented in Python3.

### Usage considerations

All indicated 2FAST2Q running times were performed on a desktop PC with a 12 core 3.7 GHz processor, and 32GB of RAM. However, 2FAST2Q runs on any up-to-date desktop or laptop. When using 2FAST2Q without mismatch search (perfect alignment only), sample processing should be in the order of seconds or minutes (after file decompression). When using the mismatch search, it is possible for 2FAST2Q analysis to take several minutes per sample. When processing more than one sample, 2FAST2Q will automatically parallelize all analyses by distributing each sample per available processor core.

2FAST2Q fast sequence mismatch search function was possible due to the use of Python numpy (*Harris et al., 2020*) and numba (*Lam, Pitrou & Seibert, 2015*) modules. An advanced and in-depth tutorial on 2FAST2Q parameters is available on GitHub and PyPI.

### 2FAST2Q algorithm

When initialized in standard feature count mode, 2FAST2Q will automatically handle all compressed or uncompressed FASTQ files, and create a hash table for all supplied sequence features. 2FAST2Q will then forward all samples for parallel processing, which can be monitored via progress bars (Fig. S1). Each FASTQ file is sequentially read, saving RAM space. The individually loaded reads are submitted to trimming based on the indicated parameters, either using a fixed position, or a dynamic search. The first assumes
the presence of a fixed feature length in the same location for all reads. The second requires one or two search sequences. When one sequence (either up or downstream) is provided, 2FAST2Q will search the read until the sequence is found, and return the predetermined sized feature (again, either up or downstream). When two sequences are used, 2FAST2Q will return any feature within the found search sequences. The location and feature length parameter can thus be ignored in this latter scenario. A sequence mismatch search can also be performed.

Following read trimming, the Phred-score corresponding to each nucleotide of the trimmed sequence is considered. If any of the scores is below the indicated parameter threshold, the read is discarded.

If the read passes quality control, alignment against the input features is finally attempted. Depending on the user input, any kind of feature alignment is performed using either mismatch search or not. By default, 2FAST2Q will always first check for a perfect match. Perfect matching uses hashing, directly comparing all features to the read sequence using hashing runtime complexity. When dealing with mismatches, 2FAST2Q will perform sequence search based on a faster custom made search algorithm. At first, all feature/search sequences are converted to their numerical binary form, subsequently reducing them to integer8 format using numpy. Sequence mismatches are counted by tracking the non-zero result positions of subtracting both sequences. 2FAST2Q mismatch search is therefore based on a Hamming distance calculation. As simple numpy constructs, arithmetic operations can be easily processed using the Python Numba module njit decorator. Therefore, all 2FAST2Q search functions are pre-compiled and effectively run at much faster speeds (Fig. S2). All read sequences searches, and features mismatch alignments are performed using this approach, allowing all search operations to run faster than standard Python code. Moreover, reads that fail to safely align, within the given parameters, to any of the provided features, are stored and used for quick hashed based comparison. The same is performed for reads that align with mismatches. By performing the much faster hashed comparison, this feature avoids the slower *de novo* mismatch search for previously seen same sequence reads. Runtime is thus decreased, paradoxically maintaining sample processing time as file size increase. "Already seen read" hashing is especially useful with datasets comprising dozens of different independent samples from the same sequencing run (see results). In this case, the generated failed/passed read hash tables for each sample are compiled and used as a seed to the next batch of samples. Each new sample thus takes advantage of the already processed reads in a previous sample, avoiding reprocessing the exact same read several times.

A Python dictionary with a class feature count is used to keep track of all found aligned sequences. When no feature file is provided (*i.e.*, when running in "Extractor+Counter" mode), all found read sequences are returned and counted. Each FASTQ file will originate a unique output file. At the end of the analysis, all samples files are compiled into a single file, which can be readily used for downstream applications.

## RESULTS

### Developing 2FAST2Q

A major goal when doing targeted (amplicon) sequencing is to know the abundance of each target within a sample. To that end, we wrote the Python-based tool called 2FAST2Q (Fig. 1). 2FAST2Q is able to efficiently extract, align, filter, and count DNA sequences from standard FASTQ files in a single step. 2FAST2Q also performs mismatch sequence searching, nucleotide Phred score quality filtering, dynamic sequence search and trimming (including double sequence search), and automatically loads and detects FASTQ (.gz compressed or not) files. The program also exists as an easy-to-use intuitive executable version for MS Windows, macOS, and Linux, requiring no installation. Alternatively, 2FAST2Q is also available as a Python3 package in the PyPI repository, and can be installed with the "pip install fast2q" command. As input, 2FAST2Q requires only a FASTQ file, and, when reference feature sequences exist (*i.e.*, sgRNAs, barcodes), a .csv file with all the lookup DNA sequences. As an output, 2FAST2Q returns an ordered .csv file with all the raw feature counts per condition, as well as quality control statistics (Figs. 1B–1C). 2FAST2Q contrasts with other current methods by being easy to setup and intuitive to use (Fig. 1A), while simultaneously maintaining advanced configuration settings such as efficient mismatched sequence searching, and quality filtering. 2FAST2Q is thus able of going beyond traditional CRISPRi experimental setups, handling any kind of feature extraction, known or unknown, from FASTQ files.

### Counting features using 2FAST2Q

An important feature of performing CRISPRi-seq or RB-seq is to obtain reliable counts of each sgRNA or barcode, for any experimental condition. When using 2FAST2Q in "counting mode", (*i.e.*, for CRISPRi-seq, or sequence barcode counting), it can be used to quickly obtain an absolute feature sequence count from FASTQ files. Moreover, it might also be of interest to extract all features existing before/in-between/after a given sequence. 2FAST2Q has an "extract and count mode" for this occasion, where the program doesn't require the input of any feature sequences, and will retrieve the count of all found read sequences. In both instances, the program can search for any feature by either specifying a starting read position, or by providing upstream and/or downstream constant search sequences. The feature length must be specified, except in the latter, where variable sized sequences can be retrieved and/or aligned to (Fig. 2).

### Benchmarking 2FAST2Q

2FAST2Q was initially benchmarked against a published CRISPRi-seq dataset comprising 479M reads dispersed over 118 FASTQ files (*Rousset et al., 2021*). In this study, *Rousset et al. (2021)* examined which genes are essential in *Escherichia coli* under different environmental conditions using CRISPRi-seq. 2FAST2Q was used to find an alignment and count the occurrence of each feature from a list of 11,629 sgRNAs across all the 118 files. When only considering perfect alignments between a feature and a read, 2FAST2Q was able to output the final compiled sgRNA count table in 7 min on a personal desktop computer (33s per sample distributed over 10 parallel cores). For comparison, the same files and parameters

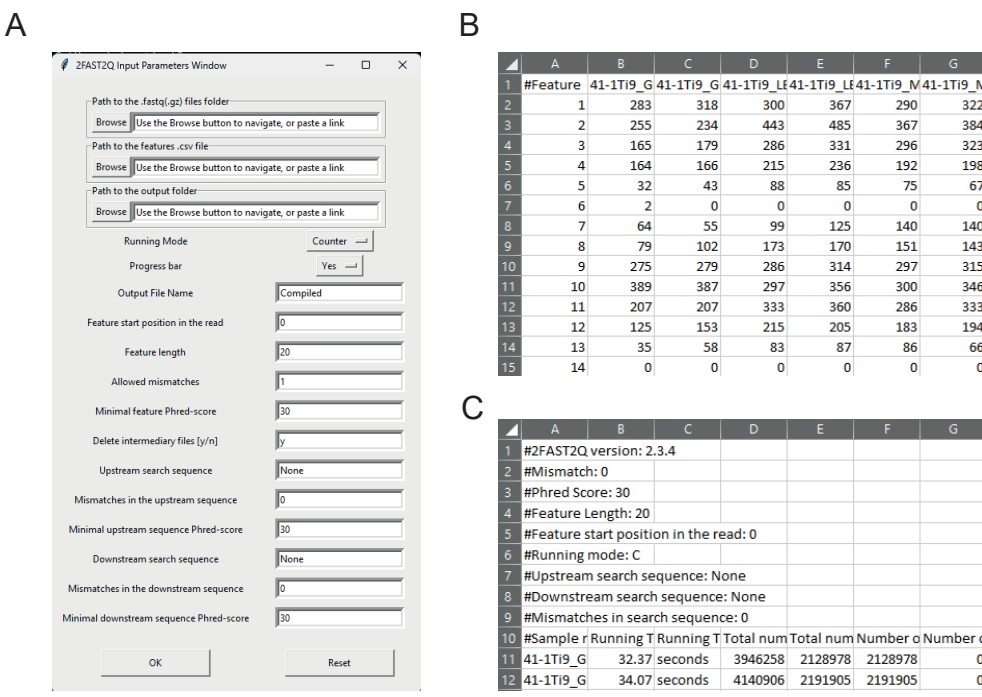

**Figure 1  2FAST2Q: A general-purpose sequence search and counting program for FASTQ file 2FAST2Q interface, and outputs.** (A) All program parameters are given by interacting with 2FAST2Q user interface. 2FAST2Q outputs two .csv files; a raw read count file for all samples (B), and a file with each sample statistics (C). Each independent file is considered to be a sample, and the file name the sample name.

were also input into MAGeCK. Despite its faster individual file processing speed, its lack of inbuilt sample multiprocessing resulted in a total run time of 23 min. Moreover, MAGeCK fails to return an organized file for all combined samples, leaving the user with the individual count files for each sample (118 in this case). MAGeCK also requires explicit indication of all the FASTQ files to be processed, a time consuming step which 2FAST2Q performs automatically, unless indicated. When comparing the read counts returned from both 2FAST2Q and MAGeCK, a perfect correlation ($r = 1$) was observed for all features (Fig. S3), indicating similar read counting accuracy.

When allowing for one mismatch in the sgRNA search count, the total run time only increased by 2min, to 9min. Under these program conditions, this corresponds to a more than 40x speed improvement over the use of similar purpose standard search functions, such as the Python regex module match function (*Python Software Foundation*). For mismatch searching, MAGeCK requires the use of Bowtie2, and respective setup, and thus was not used for further benchmarking.

Using the same dataset published by *Rousset et al. (2021)* as benchmark data, we assessed the impact of different initial 2FAST2Q parameters on both absolute feature counts, and on downstream data analysis. When not using any Phred-score filtering ($Q \geq 0$), and not allowing for any mismatches, we were able to fully recapitulate the reported total read

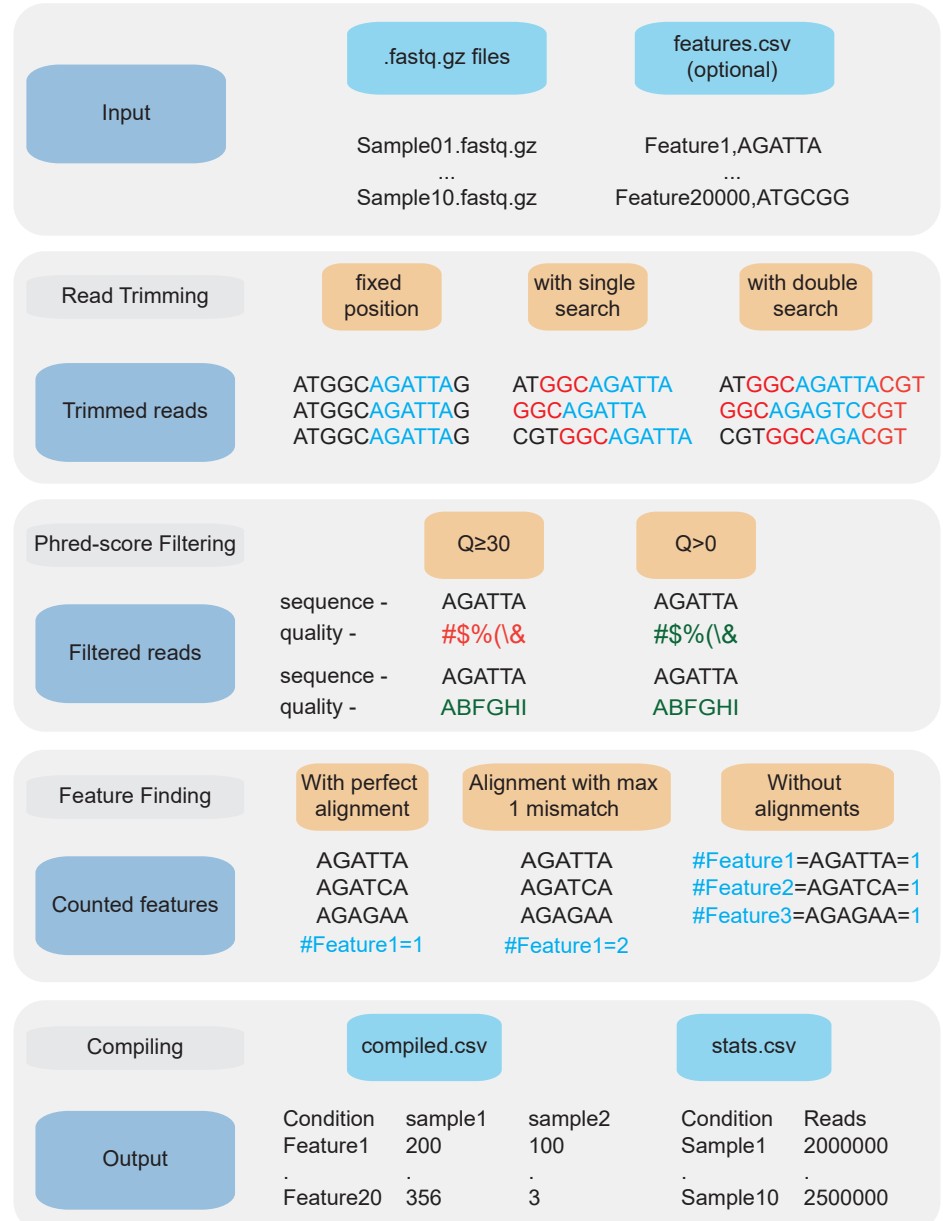

**Figure 2 2FAST2Q: A general-purpose sequence search and counting program for FASTQ files 2FAST2Q pipeline.** 2FAST2Q requires only .fastq.gz (or .fastq) files as input. When in alignment mode, a csv file with all the features must also be provided. 2FAST2Q performs all described steps automatically and without requiring external software. Trimming parameters, filtering scores, and mismatch tolerances can be easily adjusted using 2FAST2Q graphic interface.

counts/sgRNA for all conditions (Fig. 3E) (Tables S1 and S2). However, high-quality read length has been reported to improve Illumina sequencing results interpretation (*Bokulich et al., 2013*). We therefore implemented a filtering for nucleotide wise Phred-scores (Q), where all the sequenced nucleotide scores corresponding to the found feature read location are required to be above an indicated threshold. As expected, filtering using Q ≥30,
indicating a 0.1% probability of a nucleotide sequencing mistake, lowers the amount of reads/sgRNA. In some cases, by more than 1 order of magnitude (Fig. 3G). However, when considering the millions of reads generated by a typical sequencing experiment, the presence of mismatches in high quality reads is a likely event (any length of 20 nucleotides with Q $\geq$30 have, at most, a 2% chance of having a mismatch: 0.001 * 20 = 0.02). We therefore implemented feature mismatch search where a read is considered valid if it unambiguously aligns to a single feature for any number of considered mismatches, thus retrieving more high quality reads, especially from lower overall quality sequencing runs. Allowing for mismatches expectedly increased the number of reads/feature (Figs. 3A, 3C, 3I and 4A), without sacrificing total run time (Fig. 4B) (Tables S3 and S4). As an extreme benchmark case, we allowed for the same number of mismatches as the feature length (20 bp) (Figs. 3I–3J). In practice, these parameters mapped any read to its closest feature, meaning the sequence that unambiguously differs the least from the read. This is performed by an inbuilt safety mechanism, where if more than one feature possible matches the read at the lowest amount of allowed mismatches (*i.e.*, 1), the read is always discarded, but otherwise kept. In regards to the *Rousset et al. (2021)* dataset, which is on average of high quality, these parameters recovered on average 3% more reads/sgRNA (Fig. 4A). However, it is conceivable that the use and outcome of these parameters varies depending on the experimental setup and user requirements, requiring careful consideration before proceeding to downstream data analysis. In here, we report only on the possibilities of 2FAST2Q functionalities.

## Higher stringency parameters can aid in biological discovery

We used the Jupyter notebook analysis pipeline published by *Rousset et al. (2021)* to assay how these different read processing scenarios impact downstream analysis. Using the different read count tables directly outputted by 2FAST2Q, we calculated and compared the median gene scores as defined by *Rousset et al. (2021)* (essentially, the median of the log$_2$ fold change for each feature in all experimental replicates) for the LB medium and gut microbiota medium (GMM) conditions. Using more stringent criteria than *Rousset et al. (2021)* (a gene is considered significant if it has an absolute gene fold change $\geq$ 4, instead of $\geq$ 3.5), we compared how different Phred-scores and mismatch filtering criteria influenced downstream analysis, namely how these criteria influence gene score calculations, and thus gene essentiality (Figs. 3B, 3D, 3F, 3H and 3J).

We observed a higher stringency for the 2FAST2Q parameters of one mismatch, and base pair quality filtering of $\geq$ 30 (Fig. 3B), with fewer genes being considered essential for any given condition with these criteria than with the criteria that recapitulate the published data (0 mismatches allowed, and no Phred-score consideration) (Fig. 3F). As expected, different read filtering criteria resulted in fold change differences, and consequently in differences in the genes considered essential for these conditions. What criteria to use would depend on the specifics of each individual experiment. The default 2FAST2Q parameters uses a Phred-score of $\geq$ 30, while allowing up to one mismatch (representing for any 20 basepair (bp) sequence, a 5% bp deviation error with, at maximum, a 2% chance of any nucleotide being wrongly sequenced). As shown in Fig. 3, although the default setting of 2FAST2Q give

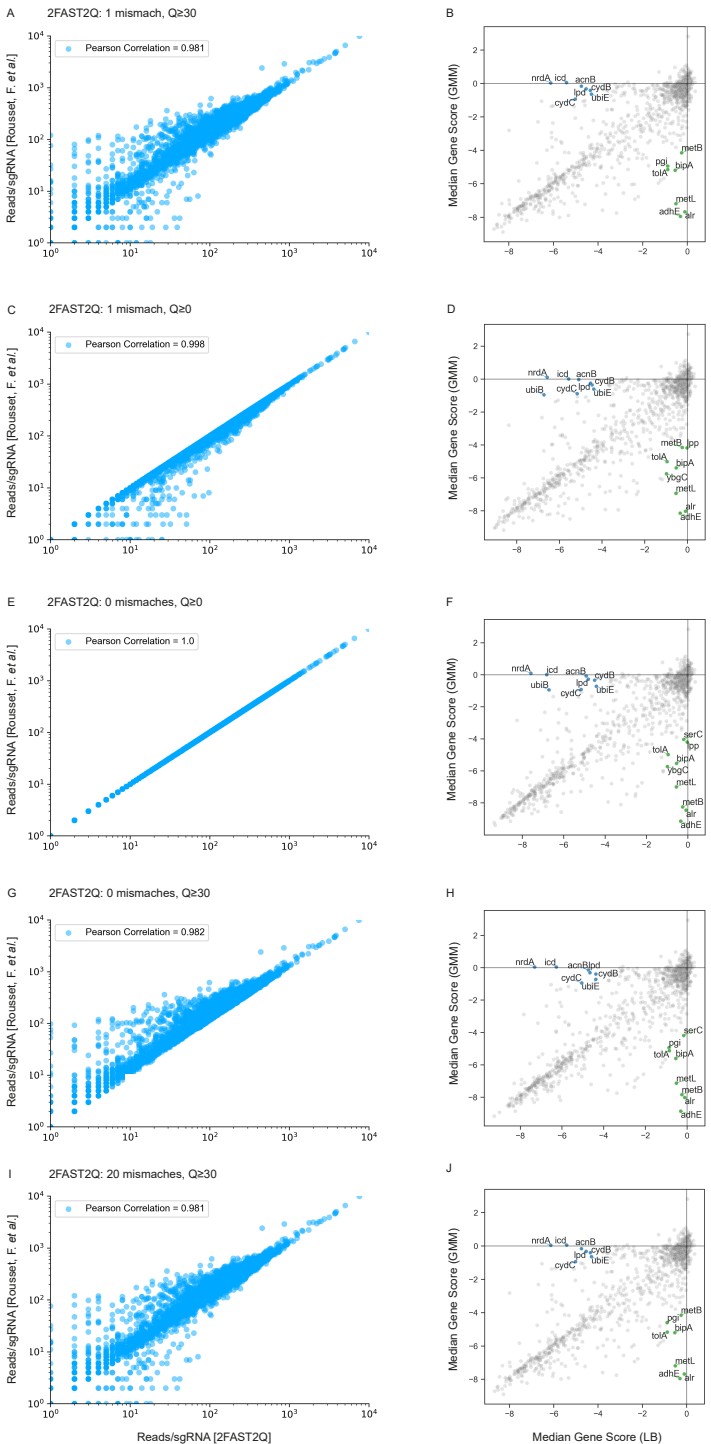

**Figure 3  2FAST2Q: A general-purpose sequence search and counting program for FASTQ file Absolute read counts/sgRNA for the _Rousset et al. (2021)_ dataset MG1655 LB 1 condition.** The total read counts using different 2FAST2Q mismatch and/or quality filtering inputs are plotted against those (continued on next page…)

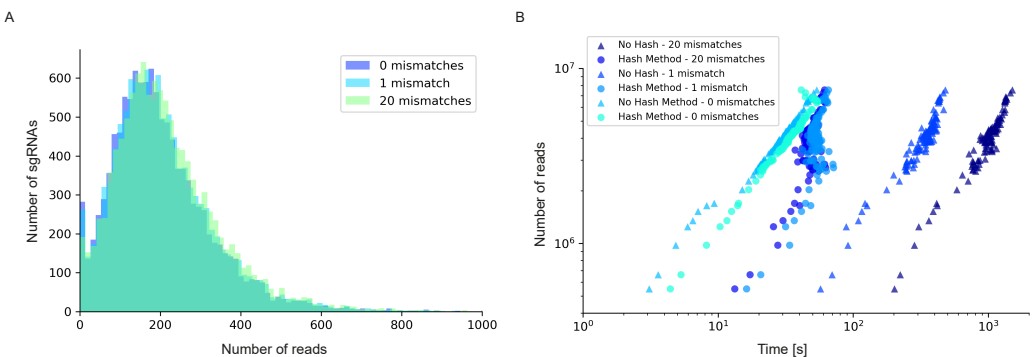

**Figure 3 (...continued)**
reported by *Rousset et al. (2021)*. Pearson correlation for each plot is also shown. Plots B, D, F, H and J were generated using an adaptation of the published Jupyter notebook analysis pipeline, and highlight the significant genes (absolute fold change ≥4 in one condition and ≤1 in the other) when using different 2FAST2Q input parameters. green: fold change <4 in GMM media, and > −1 in LB; blue: fold change <4 in LB media, and >−1 in gut microbiota medium (GMM).

**Figure 4** **Read/sgRNA distribution and runtime analysis of 2FAST2Q with different mismatch parameters and algorithms.** Data analysis was performed on the *Rousset et al. (2021)* "UTI89_T0" fastq sample (*Rousset et al., 2021*) when submitted to 2FAST2Q analysis with either 0, 1, or 20 mismatches (and Phred score ≥30). Increasing mismatches allows for greater read recovery by matching a given read to its closest matching (and thus most likely) feature. (A) The median reads/sgRNA increased from 182 to 187, and then to 196, when considering 0,1, and 20 mismatches, respectively. (B) 2FAST2Q runtime analysis demonstrates the efficiency of real time creation of pre-processed failed/passed read hash tables (see methods) vs. the "no hash" method, where each read is always processed *de novo* for mismatches.

slightly fewer significant hits, they were all also reported by *Rousset et al. (2021)*. It is also conceivable for a user to be interested in aligning all reads to their closest matching feature. This is possible by setting the total amount of mismatches to the same length of the feature. In this case, the 2FAST2Q inbuilt alignment safety mechanism will prevent ambiguous read alignments from being considered. Once again, we intend only to demonstrate the range of uses of 2FAST2Q. Ultimately, the biological relevance of which parameters to choose is left upon the user.

## 2FAST2Q dynamically performs FASTQ feature extraction

Under certain experimental setups, the extraction of features from FASTQ files might require the use of a dynamic trimming and search function (*i.e.*, when the location and/or size of the feature differs from read to read) (Fig. 2). In this case, a delimiting search sequence of any length (up and/or downstream of the feature) can be provided. Similar to feature mismatch search, an arbitrary number of mismatches can also be indicated for the search sequence-based trimming, as well as a minimum Phred-score. 2FAST2Q will search each read for the indicated sequences, returning the correctly trimmed read for further processing, and bypassing the need for more complex tools such as Trimmomatic and Bowtie2. As a proof of concept, we used a published CRISPRi-seq dataset by *Wei et al. (2021)*, where dynamic read trimming was required. In this study, a CRISPRi screen

was performed using Vero-E6 cells (kidney epithelial cells from an African green monkey) infected with SARS-CoV-2 to identify host genes important for viral replication (*Wei et al., 2021*). In this dataset, the location of each feature was at a variable location within the read. 2FAST2Q dynamic trimming allowed each read to be independently trimmed based on the relative location of the found search sequences, thus always returning the correct feature location. Using this method, we submitted six FASTQ files (SRR14668185 - SRR14668190) for 2FAST2Q processing. As search sequence we used a 10bp upstream constant sequence (CGAAACACCG), allowing for one mismatch search error in this sequence. We used the provided list of 84,953 sgRNA sequence features, and ran 2FAST2Q (Q ≥ 30, 0mismatches). 2FAST2Q simultaneously processed all six samples, comprising 324M reads, within 8 min on a standard desktop PC (Tables S5 and S6). This result corresponds to a slowdown of only 22% (speed comparisons were determined using processed reads/second) when compared with the non-dynamic feature extraction process, such as the one we used for the same parameter 2FAST2Q run with the *Rousset et al. (2021)* dataset.

Recently, *Bosch et al. (2021)* published a CRISPRi-seq experimental setup with variable length sgRNAs. In this case, both the trimming of each read and the length of each sgRNA need to be considered read by read. This is a feature, to our knowledge, beyond easy implementation in any of the programs mentioned in this work. Once again, 2FAST2Q was also able to extract, count, and align all the found features in a *Mycobacterium tuberculosis* dataset (SRR13734827), to the provided 96,700 long sgRNA file, albeit using 2 delimiting constant search sequences (upstream: GTACAAAAAC; downstream: TCCCAGATTA), while allowing for one mismatch in each. The returned variable length sequence between the two constant search sequences was used for perfect match alignment against the sgRNAs (Tables S7 and S8). When compared with the non-dynamic extraction process, a slowdown of 44% was observed, in line with what was observed for the Wei et al. dataset.

As the sequence search algorithm uses a similar process to the one used for feature alignment mismatch, a similar speed improvement over standard Python functions is also obtained. Together, these benchmarks demonstrate that 2FAST2Q is a versatile and quick computational tool that can extract relevant features and counts from FASTQ files.

## DISCUSSION

FASTQ files are the current standard sequencing output file format. Considering that new sequencing based differential analysis techniques emerge on an almost weekly basis, the need for easy-to-use, versatile and efficient programs specifically designed for extracting and counting features form FASTQ files is pressing. To this end we have developed a fast, intuitive, and easy to use tool for counting sequence occurrences in FASTQ files. We have recently implemented 2FAST2Q in our CRISPRi-seq pipeline and have found it useful in the first step of data analysis (*De Bakker et al., 2022*). In here we describe novel 2FAST2Q functionalities and explore the program's parameter versatility, covering most current user applications that require the extraction and counting of specific feature sequences, such as CRISPRi-seq and RB-Seq. Despite only handling single-ended FASTQ files at the moment, the processing of paired-ended files is possible by running two separate

instances of 2FAST2Q. The program will automatically compile all samples at the end if all intermediary files of the first run are copied to the output folder of the second instance while processing. If both reads from the paired-ended are to be analyzed as a single contiguous read, a pre-process step of read merging (for example using PEAR) is recommended. The resulting merged reads can be input into 2FAST2Q as normal.

Depending on the desired output, current methods might require users to handle several different software pipelines in order to extract relevant data from FASTQ files. However, 2FAST2Q is a standalone program that can, in a single step, efficiently and quickly perform nucleotide wise quality filtering, mismatch sequence searching, *de novo* feature extraction, and sequence occurrence counting. 2FAST2Q outputs an individually compiled, easy to interpret, excel readable .csv file with all the feature counts per sample, alongside a file with relevant sample statistics.

2FAST2Q fully recapitulated the feature counts independently returned by MAGeCK, and reported by *Rousset et al. (2021)* for all conditions when using the same filtering criteria. 2FAST2Q was also successful at extracting features starting at different positions per read when using a published dataset of a CRISPRi screen on eukaryotic cells that were infected with SARS-CoV-2 (*Wei et al., 2021*). 2FAST2Q inbuilt search functions also allow for more complex experimental setups. For example, recent work by Bosch et al. applied CRISPRi-seq with variable length sgRNAs to identify conditionally essential genes in *M. tuberculosis* (*Bosch et al., 2021*). By providing up and downstream search sequences, 2FAST2Q was able to extract these sgRNAs. In the case of experiments with more than one feature per read, such as with dual barcode sequencing, or dual CRISPRi-seq, it is conceivable that 2FAST2Q could also be used, taking into account that the parameters need to be adjusted to capture different features per read each time, and by compiling the data at the end.

Besides being able to align and count provided features in FASTQ files, 2FAST2Q is also able to extract and count all unique read sequences when in "extract and count mode". In this case, all different sequences that fulfill the required parameters are returned, with any possible mismatches being accounted as distinct sequences.

As experiments that produce large datasets (>1GB) become more widespread, the need for versatile, fast and easy to use software that handles raw data becomes more pressing. It is thus our hope that 2FAST2Q can contribute to facilitate the processing of the large amounts of sequencing data originating from NGS studies.

## CONCLUSIONS

Here, we explored and benchmarked 2FAST2Q, a tunable novel Python3-based program capable of single-step quality filtering, read feature searching, extraction, and feature counting in FASTQ files. 2FAST2Q exists as a standalone program, not requiring any installation whatsoever, and as a Python module available at the PyPI depository. We demonstrated how 2FAST2Q can be used for the processing of FASTQ files originating from different experimental setups, and how it handles different input parameters to adapt to most conceivable datasets requiring feature counting. 2FAST2Q is an easy to use

program, that we believe can streamline sequencing data feature extraction for most users, without the need for advanced bioinformatics setups, or the use of multi-step complex pipelines.

## ACKNOWLEDGEMENTS

We thank Julien Dénéréaz and Vincent de Bakker for their software tests, and all members of the Veening lab for helpful discussions.

### Funding
Work in the Veening lab is supported by the Swiss National Science Foundation (SNSF) (project grant 310030_192517), SNSF JPIAMR grant (40AR40_185533), SNSF NCCR 'AntiResist' (51NF40_180541) and ERC consolidator grant 771534-PneumoCaTChER. The funders had no role in study design, data collection and analysis, decision to publish, or preparation of the manuscript.

### Grant Disclosures
The following grant information was disclosed by the authors:
Swiss National Science Foundation (SNSF): 310030_192517.
SNSF JPIAMR: 40AR40_185533.
SNSF NCCR 'AntiResist': 51NF40_180541.
ERC consolidator: 771534-PneumoCaTChER.

### Competing Interests
The authors declare there are no competing interests.

### Author Contributions

- Afonso M. Bravo conceived and designed the experiments, performed the experiments, analyzed the data, prepared figures and/or tables, authored or reviewed drafts of the article, and approved the final draft.
- Athanasios Typas conceived and designed the experiments, authored or reviewed drafts of the article, and approved the final draft.
- Jan-Willem Veening conceived and designed the experiments, authored or reviewed drafts of the article, and approved the final draft.

### DNA Deposition
The following information was supplied regarding the deposition of DNA sequences:

All the datasets analysed during the current study are available in the European Archive Depository (ENA).

The Rousset, F. et al. dataset is available at: PRJEB37847. The Wei et al. datasets are available at: SRR14668185, SRR14668186, SRR14668187, SRR14668187, SRR14668187, and SRR14668190. The Bosch et al. dataset is available at: SRR13734827.

## Data Availability

The program is available as a standalone executable on MSwindows and MacOS, and Zenodo: afombravo, & Veening lab. (2022). veeninglab/2FAST2Q: V2.5.0 (V2.5.0). Zenodo. https://doi.org/10.5281/zenodo.7012813.

2FAST2Q is available as a python package (https://pypi.org/project/fast2q/) and is also available at Github: https://github.com/veeninglab/2FAST2Q.

## Supplemental Information

Supplemental information for this article can be found online at http://dx.doi.org/10.7717/peerj.14041#supplemental-information.

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
