# Peer review of "FAST2Q: a general-purpose sequence search and counting program for FASTQ files"

_PeerJ, doi:10.7717/peerj.14041_

## Round 0.1 · original submission · Major Revisions

Dear Dr. Bravo and colleagues:

Thanks for submitting your manuscript to PeerJ. I have now received three independent reviews of your work, and as you will see, the reviewers raised some concerns about the research. Despite this, these reviewers are optimistic about your work and the potential impact it will have on research studying genomics and bioinformatics. Thus, I encourage you to revise your manuscript, accordingly, taking into account all of the concerns raised by both reviewers.

There are many suggestions, which I am sure will greatly improve your manuscript once addressed.

I look forward to seeing your revision, and thanks again for submitting your work to PeerJ.

Good luck with your revision,

-joe

·

Basic reporting

This manuscript reports in detail the functionality and potential applications of 2FAST2Q, a program built in Python to extract and count feature occurrences in new-generation sequencing (NGS) FASTQ files. For most researchers currently working on NGS (esp. Illumina) data, there are several steps in bioinformatic pipelines that are covered by this program, taking the FASTQ files through read trimming, quality control, feature search/alignment and counting.

As stated by the authors, these represent substantial time savings and may help expand the use of NGS. To my knowledge, there has not been a software much like this one, but I note that it is also a relatively specific routine. As experiments in various fields increase in the use of NGS, this routine may be in higher demand. I certainly hope this program will help shape experiments to more easily generate and analyse NGS data.

However, one important description missing from the manuscript concerns the specific steps that typical bioinformatic pipelines require users to go through. Describing what these are—particularly the more complex or time-consuming parts of the typical pipeline—would encourage more users to adopt this program.

There are also many non-Illumina sequencing platforms today. Can it handle other sequencing read types? It would be good to mention what are the upstream processes needed by the user to convert outputs to a FASTQ form that can be read in 2FAST2Q. For example, FAST5 from ONT nanopore sequencing needs to be base-called to FASTQ. What about others if compatible?

Experimental design

The benchmarking analyses based on e.g. Rousset et al. (2021), Wei et al. (2021), Bosch et al. (2021) are straightforward and report what are needed to validate the accuracy and efficiency of the program.

Validity of the findings

The benchmarking results appear valid and supported.

Additional comments

Line 160–163: These lines describe the FASTQ format and should probably be at the start of the Materials & Methods instead of Results. And Illumina should be word-capitalised.

Reviewer 2 ·

Basic reporting

The structure of the article is in acceptable format and structure. The body of the article is written in clear and professional English, but some typing errors are still present. Moreover, the paper includes a sufficient introduction and background. However, references to other state-of-art software along with a clear specification of relevant advantages of the present work are missing.

Reported figures are in sufficient resolution and appropriately described. Besides, benchmark data and software installation are reported appropriately and are already publicly available.

The submission is not self-contained as some results relevant to the hypothesis are not reported.

Experimental design

The article is in in line with Aim and Scope of the PeerJ Journal and the submission clearly defines the research question and the knowledge gap. In addition, it is specified how the development of this program should contribute to filling that gap. The investigation has not been rigorously conducted since innovation of the program compared to the state-of-art has not been demonstrated through results (see Validity of Findings section).

Methods have been described with sufficient information to be reproducible. However, I would suggest including a more descriptive introduction to the materials of the study like type and approach of sequencing and description of the analyzed samples.

Validity of the findings

The impact and novelty of this research has been partially limited since the software has not been properly compared with other state of art features counting programs, and the accuracy of the algorithm has not been evaluated. I strongly recommend evaluating then concordance of the software with other software used in literature (see General Comments section).

The data (program code and benchmark data) is already public.

Finally, the conclusions are appropriately stated and connected to the original question but lack of reference to the state-of-art.

Additional comments

The author tested the performance of 2FASTQ on Rousset et al. experimental datasets where CRISPR inference and RNA sequencing has been applied on 370 E. coli genomes to determine the presence of genes. In this study, 2FASTQ was applied as internal step of CRISPRi-seq, CRISPR interference platform. In the present work, the author assessed the concordance 2FAST2Q results using different parameters on feature counts presented in Rousset et al. experimental dataset. To improve the paper, I would suggest performing a more in-depth benchmarking analysis considering more experimental datasets where 2FASTQ has not been applied. In addition, I suggest demonstrating the performance of 2FASTQ in comparison with other software present in literature which automatically perform read trimming and counting designed for CRISP experiments (for instance MAGeCK, CRISPRroots, CRISPRMatch, CRISPRAnalyzeR, PinAPL-Py, caRpools) or custom bioinformatic pipeline involving different software (for example cutadapt+bowtie+MAGeCK iNC).

The paragraph “Developing 2FASTQ” (line176-179) described in the Results between 160 and 179 lines is focused on the utility of the software and its applications. I believe that this part is more useful in the introduction since it apports general information. Moreover, I recommend that the authors should emphasize why 2FASTQ tool is different from any software present in literature (see software mentioned above)

In order to have more visibility, I suggest that the authors indicate in the abstract that 2FASTQ is a program suitable for targeted sequencing datasets.

In addition, I would recommend removing the software code reported between line 203 and 204. I notify that a typing error is present in line 140.

Finally, in the article, the expression “large dataset” used in lines 101,146,341 is not specific and the amount of sequencing data should be specified.

·

Basic reporting

The manuscript presents 2FAST2Q, a general-purpose sequence search and counting program for FASTQ files. 2FAST2Q provides a standalone python command-line interface and also a graphical user interface; statically-compiled executable binary files for Linux/Windows/MacOS are also available for wider users. The scenarios of 2FAST2Q are mainly short-read amplicon sequencing, e.g., CRISPRi-Seq, Tn-Seq, and RB-Seq. 2FAST2Q extracts 1) the target subsequence of a certain length at a fixed position or 2) that of fixed or variable length located next to certain upstream/downstream sequences.

Experimental design

2FAST2Q is implemented as a single Python script of about 1,000 lines. Most functions are well annotated and easy to read. The installation is simple and fast, and it did take too much time to learn how to use the CLI or GUI. The algorithm is simple and fast and is well explained.

The authors assessed the impact of different 2FAST2Q parameters (Phred-score, mismatch) on runtime, absolute feature counts, and downstream data analysis using a CRISPRi-Seq dataset, while the analysis mode is for the target sequence of a fixed length at a fixed position.

For dynamically feature extraction, only speed was compared with the non-dynamic feature extraction process. Actually, for these kinds of analyses, a slowdown of 22% or 44% is acceptable, while the accuracy is more important. For the Rousset, F. et al. Dataset [10], the author used a 10bp upstream constant sequence (CGAAACACCG) for sequence search. As I understood, the non-dynamic feature extraction can also be applied here, this should be where the 22% came from.

Validity of the findings

I simply tested the CLI with example data provided in Github and compared the result with my tool. The results were nearly identical except only the first appeared one of the signatures with the same sequence was reported in 2FAST2Q, which was also mentioned in the CLI log. For the non-dynamic feature extraction process, the locations of target subsequences need to be identical across all reads, but is this true in real data? Will sequencing error, adapter sequence, and quality control process affect the locations?

Similarly, for dynamically feature extraction, the downstream sequence would be incomplete after QC. And was merging of paired-end reads needed and how will it affect the results?

Additional comments

Major:

1. A bug in searching with upstream/downstream sequence. 2FAST2Q.py L213 (e6e3495), “reading[1]” should be converted to “str” type, which is required by “bytearray” in L285.

2. About Fig2. The decompressing part is unnecessary and misleading because 2FAST2Q accepts and reads .gz files immediately without generating intermediate .fastq files. As for Feature Finding, it would be better to highlight the matched feature sequence(s) and use “# Feature1=1” instead of “Feature1=1”, and “With 1 imperfect alignment” may be changed to “Alignment with max 1 mismatch”. Another question, is quality filtering applied to the downstream search sequence?

3. When running the graphical user interface, the paths of the input and output folder had to be configured by manually clicking and choosing via the “Browse” button, and typing in the path will make the GUI crash. And the Tkinter file browser does not save the previous status, making it verbose to repeatedly choose long and deep paths. It would be more friendly if parameter checking is done within the GUI.

Minor:

1. The abbreviation “JIT” is not clear for non-developer readers. And it only appears once in the Abstract.
2. L140 typo: ]align
3. L165 invalid citation:“(ref de Bakker)”
4. L170 “easy to use” → “easy-to-use”
5. L173 typo? “a FASTQ /gz file”
6. Typo in source code: lenght → length
7. A suggestion: do not save timestamp in the output directory name, which results in too many folders after trial and error.

---

## Round 0.2 · accepted · Accept

Dear Dr. Bravo and colleagues:

Thanks for revising your manuscript based on the concerns raised by the reviewers. I now believe that your manuscript is suitable for publication. Congratulations! I look forward to seeing this work in print, and I anticipate it being an important resource for groups studying genomics and bioinformatics. Thanks again for choosing PeerJ to publish such important work.

Best,

-joe